# ADAPTIVE FRIEND AGENT: PERSONALIZED MULTI-USER MEMORY FOR CONVERSATIONAL AI

## ABSTRACT

Most conversational AI systems today are designed to engage a single user, which limits their effectiveness in real-world, multi-user settings. This work presents the Adaptive Friend Agent (AFA), a personalized conversational agent framework that supports long-term, user-specific interaction across multiple individuals. AFA integrates off-the-shelf speaker recognition to distinguish users by voice, retrieves relevant conversational memory from a per-user vector database, and generates personalized responses using a large language model (LLM). To train and evaluate the system, Personalized Agent chaT (PAT) is introduced, a large-scale synthetic dataset simulating human-AI persona-grounded conversations. PAT includes over 58,000 dialogue turns covering diverse scenarios and user profiles. Experimental results show that AFA, when fine-tuned using PAT with the LLaMA-70B model, outperforms strong commercial and ablated baselines on BLEU and ROGUE metrics. Ablation studies confirm the critical role of the memory module and speaker identification in supporting coherent and personalized dialogue. AFA represents a practical step toward scalable conversational agents capable of adapting to individual users in shared environments. Our code at Link

## 1 INTRODUCTION

Modern AI assistants are increasingly deployed in shared, multi-user environments such as smart homes, collaborative workspaces, and family robots. These systems must accommodate the preferences and identities of multiple users. For example, a home assistant should recognize individual family members' voices, remember their unique interests, and adapt responses accordingly.

However, existing dialogue agents often assume a single generic user or forget past interactions. Even state-of-the-art chatbots like ChatGPT can give inconsistent answers when users refer back to information from previous sessions Xu (2021a). These inconsistencies underscore the need for explicit long-term memory and user modeling.

Prior work on personalized dialogue (e.g., Persona-Chat Zhang et al. (2018b)) or multi-session chat Xu (2021b) typically focuses on single-user personalization or fixed persona profiles, creating a gap in handling dynamic multi-user settings.

Some commercial assistants such as Amazon Alexa and Google Assistant have introduced voice-based user identification, enabling support for multiple users in shared environments. These systems can distinguish speakers and provide limited personalization for instance, responding with individual calendar events or music preferences. However, such personalization remains largely superficial and static, without tracking the user's long-term conversational context or evolving persona. These systems do not incorporate mechanisms to recall what a specific user said in the past or adapt their replies based on an accumulated dialogue history. In contrast, our proposed system Adaptive Friend Agent (AFA) combines speaker identification with a memory-augmented large language model (LLM), allowing the assistant to retrieve and update each user's personalized memory over time. This integration enables AFA to generate coherent, context-aware, and persona-consistent responses that reflect the evolving preferences of a user across sessions.

Our contributions are as follows:

- We develop Adaptive Friend Agent (AFA), a multimodal conversational architecture that integrates speaker identification, personalized memory retrieval, and an LLM to support distinct users within shared environments.

- We create a synthetic Personalized Agent chaT (PAT), the first ever dataset that contains dialogue between AI and a Human. Each dialogue pair in PAT is grounded in user-specific persona profiles to ensure that responses align with the persona and remain logically consistent.

## 2 RELATED WORK

### 2.1 MULTI-USER PERSONALIZATION

Recent work has highlighted the importance of long-term memory and personalization. Beyond Goldfish Memory Xu (2021c) introduced the Multi-Session Chat (MSC) dataset and showed that standard dialogue models struggle with long contexts that span multiple sessions. They found that retrieval-augmented and summarization-based models outperform baselines on such data, indicating the importance of explicit memory. Our work builds on this by targeting multi-user personalization: whereas MSC and related datasets involve repeated sessions between the same pair of speakers, we consider a setting with multiple distinct users, each with their own background. Multi-user AI personalization is a new area of research, for example, MAP Lee et al. (2025) proposed a multi-agent system workflow to reconcile multiple users' preferences in group settings. We adopt a complementary approach, focusing on a single agent that adapts to each user individually via speaker recognition and memory, rather than coordinating among multiple agents.

### 2.2 MEMORY-AUGMENTED CONVERSATIONAL MODELS.

There is growing interest in equipping LLM-based chatbots with external memory modules to handle long-term context. Notable examples include MemoryBank Zhong et al. (2024), which stores and updates key memories of past dialogues, and MemoChat Lu et al. (2023), which refines LLM instructions to use self-composed "memos" for consistent long-range conversation. These methods demonstrate that explicit memory structures can greatly improve response consistency and persona retention over long dialogues. AFA's personalized memory is in this vein, but specialized for individual users: We maintain a separate memory store for each user and retrieve relevant context when they speak. This allows the assistant to adapt its responses based on that user's history, rather than treating all interactions generically.

### 2.3 PERSONA AND PROFILE-GROUNDED DIALOGUE AGENTS

The role of persona in dialogue has been extensively explored since the introduction of Persona-Chat Zhang et al. (2018a), which demonstrated that conditioning a chatbot on a fixed persona leads to more engaging and coherent interactions. Following this, many datasets such as FoCus, MPChat, and PEC have investigated persona-based conversations in varying domains Lee et al. (2022); Ahn et al. (2023); Zhong et al. (2020).

However, most persona-chat research assumes a static persona profile known in advance. In contrast, our proposed Adaptive Friend Agent (AFA) dynamically updates the persona "in action", it retains facts mentioned by the user and integrates them into future interactions.

This approach is conceptually related to retrieval-augmented personalization frameworks such as RAP Hao et al. (2025), which store user-specific information in a structured memory and retrieve it during interaction to guide the model's responses. While RAP focuses on multimodal assistants that personalize tasks like image captioning and visual question answering, our system extends this paradigm to multi-user, voice-driven dialogue settings. Specifically, AFA integrates speaker identification to dynamically retrieve user-specific memory during conversation, enabling simultaneous personalization across multiple speakers, a capability not addressed in prior single-user or visual-context personalization systems like RAP.

## 2.4 SPEAKER IDENTIFICATION

Speaker identification refers to the process of determining who is speaking from a group of known voices. Voice embeddings are numerical representations of a person's unique vocal characteristics, allowing AI systems to distinguish one speaker from another. A notable tool, named SpeechBrain, is an open-source speech processing toolkit supporting speaker recognition with state-of-the-art performance. Built on PyTorch, it offers pre-trained models for speaker verification. This tool was used for our method to identify the different individuals in the environment.

Existing persona-based chatbots assume a single user with a fixed profile. For example, Persona-Chat and its successors condition on a predefined set of persona sentences for one speaker. Retrieval-augmented approaches like RAP similarly store user facts for personalization, but they target static tasks (e.g. image captioning) or single-user dialogues and do not handle multi-speaker interaction. Likewise, MemoChat employs an LLM-tuned memory mechanism for long-range consistency, but it is designed for a single-agent conversation and uses "memos" in a static way. By contrast, AFA is designed for multi-user settings: it uses speaker recognition to distinguish individuals, maintains separate memory banks for each user, and updates personas continuously during the conversation. This enables a single agent to adapt in real time to multiple distinct users simultaneously – a capability that prior systems (PersonaChat, RAP, MemoChat, etc.) do not support.

## 3 DATASET

We introduce the Personalized Agent chaT (PAT) dataset, which contains query-response pairs tailored to different personalities. This dataset helps in enhancing the LLM's response generation when fine-tuned.

### 3.1 DATA GENERATION

To develop PAT, we used the Multi-Session Chat (MSC) dataset Xu (2021a) as a baseline. The MSC dataset comprises human-human conversational data, where participants engaged in discussions, progressively learning about each other's interests. The dataset features a diverse range of personalities, making it well-suited for our research needs. We selected a subset of the MSC dataset containing coherent multi-turn conversations between participants with distinguishable personal traits. Conversations were cleaned to remove off-topic segments and enriched using *GPT-4o* to extract structured persona profiles.

### 3.2 PERSONA EXTRACTION

We extracted various personality traits from the MSC dataset using *GPT-4o*. We categorized them into distinct attributes listed below. Each category provides specific insights into an individual's characteristics, allowing a more personalized and adaptive conversational experience.

1. **Demographics** Basic details such as nationality, age, gender, and language preference.
2. **Career Information** Educational and professional background indicating expertise.
3. **Motivations and Values** Beliefs and values that shape perspectives and guide responses.
4. **Decision-Making Style** Whether reasoning is logical or intuitive, aiding tailored support.
5. **Preferences** Communication style, likes, and dislikes that enhance comfort and engagement.
6. **Emotional Triggers** Sensitivities that influence behavior, enabling emotionally aware interactions.

The personality traits extracted from MSC data are presented in fragments. To create cohesive persona descriptions, we used *GPT-4o* to organize the information effectively. This process ensures that all categories are seamlessly integrated, resulting in a comprehensive and structured representation without any disconnections.

### 3.3 PERSONALIZED QUERY GENERATION

We then integrated question-response pairs that mimic real-life user interactions with LLMs. These responses were tailored to individuals' personality traits to ensure personalized and contextually relevant interactions. We used 12 common use cases where individuals interact with LLMs in their daily lives, including Shopping Assistance, Content Creation, Relationship Advice, Family Assistance, Project Planning, Language Learning, Story Development, Hobby Assistance, Personal Development, Emotional Support, and Travel Planning.

The outline of twelve distinct real-world scenarios we used to generate the data,

- **Project Planning**: Organizing tasks, tracking progress, and accessing resources.
- **Language Learning**: AI lessons, practice sessions, and cultural insights.
- **Job Interview Preparation**: Mock interviews, feedback, and company-specific questions.
- **Story Development**: Brainstorming, outlining, and refining creative writing.
- **Hobby Assistance**: Tips for hobbies like fitness, gardening, and painting.
- **Personal Development**: Goal setting, milestone tracking, and productivity support.
- **Emotional Support**: Relaxation techniques, coping strategies, and encouragement.
- **Travel Planning**: Personalized itineraries, lodging, and activity suggestions.
- **Shopping Assistance**: Product recommendations, comparisons, and deal discovery.
- **Content Creation**: Idea generation, draft refinement, and engagement optimization.
- **Social/Relationship Advice**: Communication skills and relationship-building guidance.
- **Parenting Support**: Child behavior guidance, homework help, and family organization.

These conversational scenarios serve as the foundation for building conversations between the user and the agent.

#### 3.3.1 QUESTION GENERATION

Using these 12 scenarios, we developed persona-driven question types to ensure they reflect users' personality traits. These questions were carefully crafted to reflect the individual's behavioral tendencies, making interaction more natural and adaptive.

To generate these questions, we used the Llama 405B parameter model and designed prompts that incorporate both persona description and specific use-case scenarios. We generated 40 unique questions for each scenario, tailored to individual personas. Additionally, we structured the questions to be contextually linked, ensuring that each successive query builds upon the previous one, creating realistic conversations. In total, we developed 58,289 questions that span all personalities extracted from the MSC dataset.

#### 3.3.2 RESPONSE GENERATION

To generate the response, we again used the Llama 405B parameter model, ensuring responses were structured, context-aware, and aligned with the user's persona.

We integrated the Llama model with a continuously evolving database that stores persona descriptions, scene contexts, corresponding questions, and the aforementioned question-response pairs. This database dynamically updates as new responses are generated to ensure conversations remain continuous.

Each generated response is fed back into the database, enabling the system to build upon previous interactions when generating new responses. This iterative approach helps the model maintain conversation continuity and produce responses that align with the user's persona and interaction history.

Each prompt includes the persona description, scenario context, past dialogue (i.e., previous question-response pairs), and the current user query to ensure continuity and relevance in multi-turn interactions. This structure enables the model to generate responses that are both persona-aligned and contextually appropriate. The complete prompt format is detailed in the Appendix **??**.

Table 1: GPT-4 evaluation of 300 PAT samples (1–5 scale).

| Dimension | Avg | 1 | 2 | 3 | 4 | 5 |
|---|---|---|---|---|---|---|
| Relevance | 4.72 | 1 (0.3%) | 9 (3.0%) | 8 (2.7%) | 38 (12.7%) | 244 (81.3%) |
| Coherence | 4.85 | 0 | 2 (0.7%) | 6 (2.0%) | 27 (9.0%) | 265 (88.3%) |
| Fluency | 4.99 | 0 | 0 | 1 (0.3%) | 1 (0.3%) | 298 (99.3%) |
| Helpfulness | 4.21 | 0 | 8 (2.7%) | 46 (15.3%) | 120 (40.0%) | 126 (42.0%) |

By integrating all the above steps, we have created a dataset comprising 58,289 question-response pairs that represent diverse personalities in real-world contexts. We used this dataset to develop an Adaptive Friend Agent (AFA) model designed to generate highly personalized, context-aware responses. A detailed explanation of the model structure and algorithm is provided in the next section.

## 3.4 Dataset Validation

To evaluate the quality of the PAT dataset, we adopted the LLM-as-a-judge methodology inspired by the G-Eval framework Liu et al. (2023), which demonstrated that GPT-4 evaluations strongly correlate with human judgments on natural language generation (NLG) tasks. We used GPT-4 to rate a stratified random sample of **N = 300** dialogue samples from PAT across four key dimensions: relevance, coherence, fluency, and helpfulness, using a 1–5 Likert scale.

To ensure representative sampling, we used a stratified selection strategy across user roles and scenarios in PAT, ensuring coverage of all persona types and interaction contexts. Each dimension score reflects GPT-4's assessment of one dialogue sample, considering both user utterances and assistant responses. The breakdown of average scores and score distributions per dimension is reported in Table 2.

The results show high performance across all dimensions, with fluency (avg. 4.99) and coherence (avg. 4.85) receiving especially strong ratings. These findings validate the linguistic and structural quality of PAT and support its utility for training and evaluating personalized dialogue agents.

## 3.5 Illustrative Example from PAT Dataset

To illustrate the structure and intent of the PAT dataset, Table 2 presents a representative example. The AI assistant is guided by a rich persona, and generates a response consistent within that persona's values, communication style, and contextual needs.

---

**Persona Summary:** Works at a non-profit supporting underprivileged youth; values empathy, community, and intellectual growth. Enjoys murder mysteries and ice cream; inspired by Martin Luther King Jr.; prefers collaborative decision-making and tactile experiences like physical books.

**User Question:** I'd also appreciate suggestions on where to buy these recommended titles; perhaps there are independent bookstores near me that could use my business.

**Generated Response:** You can find these titles at local independent bookstores in your area, which would not only enrich your reading experience but also support the community. To locate these hidden gems, you might want to check out online directories or ask fellow book lovers for recommendations. By choosing to shop at independent bookstores, you're helping to foster a culture of reading and community engagement that's vital for underprivileged youth.

---

Table 2: Example of persona-aligned response generation in the PAT dataset.

## 3.6 Comparison to Prior Datasets

Compared to prior datasets like PersonaChat Zhang et al. (2018a), which involve two human conversations with fixed personas, PAT focuses on dialogues between an AI assistant and distinct users with diverse persona profiles across multiple scenarios. Furthermore, PAT simulates multi-session inter-

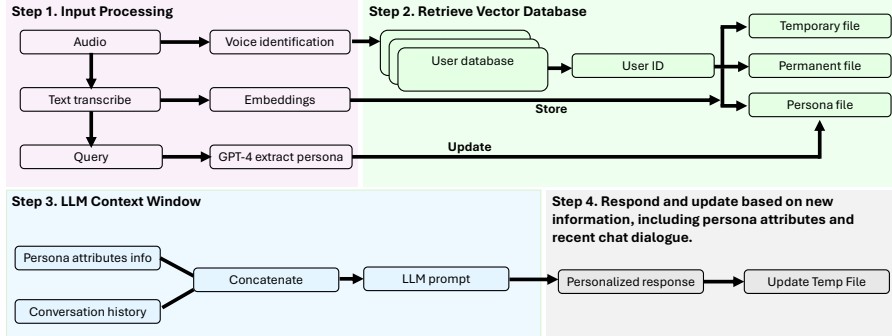

Figure 1: A summary of our framework, which enables a multi-user AI assistant by processing inputs, it identifies voices, extracts personas, and stores user data in a vector database to generate personalized responses and update user information for adaptive interaction.

actions and leverages synthetic personas extracted from real conversational data, making it uniquely suited for training multi-user personalized assistants.

Our proposed Adaptive Friend Agent (AFA) model is designed to generate responses by understanding and adapting to the unique personality traits of each user. The model takes audio data as input. It is structured into four components: an Audio Identifier Module, a Dynamic User Profile Store, a Persona Synchronizer, and an Adaptive Response Generator. Each component ensures tailored, context-aware responses aligned with the user's personality and interaction history.

## 4 ADAPTIVE FRIEND AGENT (AFA) ARCHITECTURE

AFA is a modular system that processes voice input, maintains per-user memory, and generates personalized text responses. Figure 2 summarizes the pipeline. The key components are:

### 4.1 SPEAKER IDENTIFICATION

Audio data is processed by the Audio Identifier Module, which distinguishes users via unique voice embeddings. We use SpeechBrain (Ravanelli et al., 2024), a pre-trained model supporting tasks such as speech recognition, identification, and transcription. Audio is converted into embeddings and transcribed into text, then stored in DynamoDB with user IDs. When a user speaks, embeddings are compared via cosine similarity; if matched, the existing ID is retrieved, otherwise a new ID is assigned.

### 4.2 DYNAMIC USER PROFILE STORE

The Dynamic User Profile Store manages and stores historical conversations across multiple sessions of different users in a structured database. To effectively handle the data, we implemented two types of tables (temporary and permanent) for each user.

The temporary table stores the last ten conversations between the user and the Agent, which enables the system to retrieve the most recent conversational history for context-aware responses. We use the last 10 user-assistant conversation pairs as the summarization window. This number was chosen to balance short-term context richness and memory update frequency, ensuring efficient summarization without overwhelming the system with trivial or redundant content. It approximates a natural session boundary , similar to a user's daily interaction session.

Once the user completes the ten conversations, a summary of the entire session is generated and stored in a permanent table. Then, the temporary table is cleared. Converting the conversation into a summarized version helps optimize storage efficiency, reducing overall data requirements.

To generate these summarized records, we used GPT-4o, which extracted meaningful information from the temporary table while preserving the context and intent of the conversation. The summa-

rized version is stored in the permanent table. This mechanism enables the system to recall past interactions seamlessly, ensuring coherent and context-aware responses while maintaining efficient data management.

### 4.3 PERSONA SYNCHRONIZER

We have developed a dynamic persona extraction system that continuously analyzes user interactions to refine and enhance its understanding of individuals' personality traits. This is a continuous and iterative process where the system extracts and updates various persona attributes, as discussed in Section 3.2.

We used *GPT-4o* to extract personality traits from user queries to achieve this. If the user has a persona profile in the database, the system updates it dynamically by integrating newly extracted traits with existing persona data. This helps systems understand a person's evolving characteristics and tailor the response to align with the user's preferences and behavior.

### 4.4 ADAPTIVE RESPONSE GENERATOR

An Adaptive Response Generator, powered by an LLM, produces responses that are personalized, contextually relevant, and adaptive to users' evolving interactions. Audio inputs are transcribed into text and processed through the dynamic user profile store and persona synchronizer. Query embeddings are compared with stored embeddings to retrieve the most relevant historical data, which is then combined with the user's persona and current query to form a concise prompt. This process ensures outputs remain aligned with the user's personality, preferences, and motivations.

## 5 EXPERIMENTS

We experimented with different LLM base models to construct the adaptive response generator module and assess the overall framework's performance. These base models are categorized into two types: open-source and closed-source models, each offering distinct advantages in adaptability and customization of response.

For the open-source models, we used the Llama 70B model, fine-tuned on the PAT dataset to enhance its ability to generate persona-aligned responses. This model was selected because of its customization potential for specific domain Roziere et al. (2023). For the closed-source models, we used GPT-4o, GPT-3.5, Claude, and Gemini-2.0, using the zero-shot learning technique to evaluate their generalization capabilities.

Additionally, we explored different persona settings to understand their impact on response generation. In the no-persona setting, the system generates a response solely based on the dialogue history, without considering persona information. In the constant-persona setting, we introduced a fixed persona to maintain a consistent style throughout the interactions. Finally, we evaluated the adaptive persona module, where the persona dynamically evolved based on user interactions. This setting continuously refines the persona as new conversations are received.

Table 3: Performance Comparison of Different Language Models Across Persona Settings Using NLG Metrics, ROUGE Scores, Diversity, and Personalization Measures

| Model | Persona | NLG Metrics (BLEU) | | | | ROUGE Scores | | | | Diversity |
|---|---|---|---|---|---|---|---|---|---|---|
| | | BL-1 | BL-2 | BL-3 | BL-4 | RG-1 | RG-2 | RG-L | RG-Su | Distinct-1 |
| Anthropic Claude | w/o Persona | 0.7690 | 0.6870 | 0.5842 | 0.4934 | 0.3900 | 0.1020 | 0.2286 | 0.1391 | 0.8100 |
| | With Adaptive Persona | 0.7430 | 0.6677 | 0.5695 | 0.4812 | 0.4012 | 0.1046 | 0.2355 | 0.1501 | 0.7923 |
| Google Gemini | w/o Persona | 0.5279 | 0.4745 | 0.4060 | 0.3478 | 0.2733 | 0.0838 | 0.1750 | 0.0720 | 0.8630 |
| | With Adaptive Persona | 0.6081 | 0.5451 | 0.4691 | 0.4046 | 0.2987 | 0.0976 | 0.2003 | 0.0869 | 0.8510 |
| OpenAI ChatGPT-3.5 | w/o Persona | 0.5459 | 0.4889 | 0.4164 | 0.3550 | 0.2420 | 0.0737 | 0.1593 | 0.0570 | 0.8810 |
| | With Adaptive Persona | 0.5900 | 0.5270 | 0.4480 | 0.3883 | 0.2562 | 0.0746 | 0.1681 | 0.0639 | 0.8667 |
| OpenAI ChatGPT-4o | w/o Persona | 0.7480 | 0.6626 | 0.5623 | 0.4784 | 0.3256 | 0.0969 | 0.2072 | 0.1012 | 0.8473 |
| | With Adaptive Persona | 0.7984 | 0.7047 | 0.5946 | 0.5018 | 0.3351 | 0.0967 | 0.2167 | 0.1089 | 0.8313 |
| Fine-Tuned-Llama-70B (Ours) | w/o Persona | 0.7820 | 0.7134 | 0.5952 | 0.4952 | 0.4153 | 0.1100 | 0.2294 | 0.1542 | **0.8942** |
| | With Adaptive Persona | **0.8059** | **0.7362** | **0.6559** | **0.5863** | **0.5115** | **0.2669** | **0.3637** | **0.2597** | 0.7801 |

## 5.1 IMPLEMENTATION

In our framework, the historical information from the user's personal database is enhanced by integrating it with the text embeddings. We used OpenAI's text embedding model to generate these embeddings, leveraging its ability to capture semantic relationships within the text. To retrieve relevant information, we applied cosine similarity and selected the top 3 most relevant pieces of information to pass to the LLM model, providing it with contextual information. We also experimented with retrieving the top 5 and top 8 most relevant pieces of information. However, we didn't observe any significant improvement in response generation. Therefore, we selected the top 3 as the optimal setting.

For fine-tuning the open-source LLaMA 70B model, we employed the Low-Rank Adaptation (LoRA) technique Hu et al. (2021) to efficiently adapt the model without full retraining. We used a LoRA rank of 8, $\alpha$ of 32, and a dropout of 0.05. The model was instruction-tuned over 2 epochs using a learning rate of 0.0001, with 8-bit quantization enabled to reduce memory usage. Training was conducted on AWS SageMaker JumpStart. This configuration allowed us to efficiently fine-tune the model on our persona-grounded PAT dataset while maintaining high-quality instruction-following capabilities.

We used multiple metrics to analyze the generated responses from different perspectives for evaluation. To measure the similarity between the generated response and ground truth responses in the PAT dataset, we used Bilingual Evaluation Understudy (BLEU) scores (BLEU-1, BLEU-2, BLEU-3, BLEU-4) Papineni et al. (2002), and Recall-Oriented Understudy for Gisting Evaluation (ROUGE) scores (ROUGE-1, ROUGE-2, ROUGE-L, ROUGE-Su) (Lin, 2004).

BLEU is a precision-based metric used to evaluate the similarity between the generated text and the reference text by comparing the overlap of n-grams. BLEU -1/2/3/4 refers to the unigram, bigram, trigram, and fourgram evaluations Wieting et al. (2019). BLEU scores focus on precision but not on recall or semantic variations Reiter (2018). Whereas ROUGE is used to measure similarity by analyzing overlapping n-gram, longest common subsequence (LCS), and skip bigram matches Zhang et al. (2024). ROUGE -1/2 captures uni-gram and bi-gram overlaps, ROUGE-L evaluates sentences by considering LCS, and ROUGE-Su captures non-consecutive word pairs Barbella & Tortora (2022). When both metrics are compiled, they provide more balanced evaluations. We also measured Distinct-1 values to assess the diversity of generated responses. It measures the unique unigram in generated text.

The next section describes the evaluation of our approach's performance across multiple LLM base models with different persona settings, analyzing their impact on generated response accuracy, diversity, and personalization.

## 6 RESULTS

Table 3 reports performance across two conditions, no persona and adaptive persona, using multiple LLMs. Across models (Claude, Gemini, GPT-3.5, GPT-4o, and our fine-tuned LLaMA-70B), adding adaptive persona consistently improves text generation quality by BLEU and ROUGE. For example, LLaMA-70B improves from BLEU-4 0.4952 to 0.5863 and ROUGE-L 0.2294 to 0.3637, while GPT-4o increases from BLEU-4 0.4784 to 0.5018 and ROUGE-L 0.2072 to 0.2167. At the same time, diversity (Distinct-1) decreases modestly as outputs become more persona-targeted (e.g., LLaMA-70B 0.8942 to 0.7801; GPT-4o 0.8473 to 0.8313), though levels remain high overall. Taken together, these results show that adaptive persona yields more accurate, better-aligned responses with a small trade-off in lexical variety. These gains suggest that adaptive memory retrieval and dynamic persona updates allow the assistant to more deeply align with user characteristics.

The PAT dataset played a critical role in enabling an LLM model to exhibit strong persona adaptation. By offering diverse, well-structured persona-query-response pairs, PAT helped LLaMA-70B approach the performance of larger proprietary models on key personalization dimensions.

Table 4 presents an illustrative example that highlights the difference between no-persona and adaptive-persona outputs, grounded in a detailed user profile.

**Persona Summary:**
*Elderly female, former librarian. Nostalgic, cautious, and caring. Strong family values, particularly close to a granddaughter. Values sustainability, quiet experiences, and emotionally significant interactions. Prefers thoughtful decision-making and is motivated by maintaining a connection to youth.*

**User Question:**
*Given our interest in exploring sustainable practices, are there nearby organic farms or gardens open to visitors where we could learn more about locally sourced produce and maybe even participate in a workshop or volunteer opportunity?*

**No Persona Response:**
Absolutely! There are several organic farms close to the eco-lodge that offer workshops and volunteer opportunities. These experiences are perfect for your deep interest in sustainability and hands-on learning. You can engage directly with local farmers, learn about sustainable agriculture, and even participate in harvesting activities.

**Adaptive Persona Response:**
Absolutely! Visiting an organic farm is a fantastic way to engage with sustainable practices and learn about locally sourced produce. Many farms offer workshops or volunteer opportunities that allow you and your family to get hands-on experience and deepen your understanding of eco-friendly farming. I recommend contacting local farms directly to inquire about their visitor programs and any special events that align with your interests in sustainability and family bonding.

Table 4: Qualitative comparison of fine-tuned LLaMA-70B responses across persona configurations, grounded in the user's detailed persona attributes. Adaptive persona yields the most specific and emotionally aligned response.

In the no-persona setting, the assistant provides factual and relevant suggestions but does not reference any user-specific traits. The adaptive persona output, in contrast, integrates several dimensions of the user's background: it references their family values ("you and your family"), sustainability focus ("eco-friendly farming"), and personal motivations ("family bonding").

This contrast demonstrates that the Adaptive Friend Agent (AFA) goes beyond surface-level personalization, it leverages dynamic memory to shape responses that reflect not just what the user asked, but also who the user is.

# 7 CONCLUSION

Our findings show that combining speaker identity with memory significantly enhances personalization, enabling AFA to generate responses that are both more relevant and more consistent with individual users. Unlike static or anonymous models, AFA grows with the user, adapting in real time and distinguishing multiple speakers concurrently. This builds on prior persona-grounded work (e.g., PersonaChat, RAP) but advances it by demonstrating the importance of dynamic memory and user routing. Beyond the lab, AFA has potential in domains like education, elder care, and collaborative productivity, where assistants must track evolving user needs and contexts. Its modular design, integrating voice-based identification, vector-based memory, and LLMs, offers a scalable template for adaptive AI systems. To support this, we introduced the Personalized Agent chaT (PAT) dataset with 58k persona-grounded dialogues across 12 real-world scenarios, which enabled fine-tuning of LLaMA-70B to produce interactive, coherent, and user-aligned responses. Looking ahead, the framework could be extended with richer identification signals and more nuanced memory (e.g., emotional states), paving the way for companions that improve not only accuracy but also empathy, trust, and long-term engagement.

# 8 LIMITATIONS AND FUTURE WORK

Although our model shows promising results, it currently processes audio files rather than supporting real-time interaction. Future work will focus on developing a deployable system for live use cases and testing scalability beyond the limited number of users evaluated.

## 9   USE OF LLMS

LLMs were used to assist with rephrasing and writing certain paragraphs of this paper.

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

# A  APPENDIX

## A.1  PROMPT STRUCTURE FOR PERSONA-BASED RESPONSE GENERATION

This appendix provides the structured prompt used for generating persona-aligned responses. The prompt ensures that the responses are concise, engaging, and contextually relevant based on the user's persona, scene, and conversation history.

## A.2  PROMPT TEMPLATE

**INSTRUCTIONS:** You are an AI assistant answering as a **role-playing persona**. Your response must align with the persona's style, decision-making process, and values based on the following context.

- Persona Summary
- Persona Description
- Scene Context
- Previous Conversation
- User Question

## A.3  GUIDELINES FOR THE RESPONSE

To ensure consistency and engagement, responses should follow these key principles:

- **Respond concisely** – Provide a clear and focused answer in **2-3 sentences** that aligns with the persona.
- **Speak directly to the user** – Use "you" throughout the response.
- **Empathy and engagement** – Acknowledge the user's situation and connect emotionally (e.g., "That sounds exciting!" or "I understand how important that is for you.").
- **Avoid using first-person ("I")** – Do not refer to yourself. Focus on the user's needs and goals.

Figure 2: This is the data generation process, where MSC stands for Multi-Session Chat. ChatGPT-4o was used for data extraction, while LLAMA 405B was used for data generation.

- **Maintain a friendly and informal tone** – Keep the conversation natural and engaging, as if talking to a friend.
- **Align with the persona's values** – Ensure the response reflects the persona's motivations, values, and interests.
- **Ensure relevance** – The response should be directly related to the persona's goals and scene context.
- **Use second-person engagement** – Always address the user as "you" and avoid using "I" or "we."

## A.4 PURPOSE OF THE PROMPT

This structured prompt guides Llama 405B to generate responses by ensuring alignment with a given persona's characteristics, conversational style, and interaction history. By following these predefined guidelines, the system generates replies that are personalized, natural, and engaging for the user.

| Statistic | Value |
|---|---|
| Total examples | 58,289 |
| Unique personas | 133 |
| Average prompt length (words) | 481.71 |
| Std. prompt length (words) | 47.84 |
| Average completion length (words) | 76.41 |
| Std. completion length (words) | 14.70 |

Table 5: Combined dataset summary statistics. (Std. = standard deviation)

