# OpenReview forum: "Adaptive Friend Agent: Personalized Multi-User Memory for Conversational AI"
_ICLR.cc/2026/Conference — Submitted to ICLR 2026_

### Official Review · Reviewer_mNQc · 2025-10-26

**Soundness:** 1
**Presentation:** 2
**Contribution:** 2
**Rating:** 2
**Confidence:** 3

**Summary:**

The authors introduce Adaptive Friend Agent (AFA) - a system for LLM conversation in a multi-user environment. They base AFA on speaker identification + speaker specific conversation history and speaker s pecific memory.

They create a synthetic dataset called PAT with (persona + user query + contextualized user response) with Llama 405B. They show that existing LLMs benefit from using persona to generate relevant responses and show that fine-tuning a LLAMA model on PAT shows the best performance on PAT compared to non-finetuned models.

**Strengths:**

- The paper covers an original field : how to adapt a LLM to a multi-user environment
- Their speaker identification + persona update/memory update / persona specific responses is well engineered and adapts well to real-world use cases

**Weaknesses:**

- claim "we create [...] the first ever dataset that contains dialogue between AI and a Human" -> this is debatable as the dataset is synthetic, there are existing human-AI interaction datasets such as OpenAssistant Conversations or Tulu SFT dataset with user/assistant pairs for LLM SFT
- the approach of using speaker identification + per-speaker history/persona is sound - but it is not compared against any baselines. The experiment/comparison is comparing existing LLMs on generating responses on the synthetic dataset (generated with LLAMA).
- The fine-tuned LLAMA model is apparently fine-tuned on the same dataset that it is evaluated on - putting the other models at a disadvantage.
- Performance on the persona+query+response benchmark is unrelated to the multi-user system that's described earlier, it is a single user task.
- The models are evaluated based on BLEU and ROUGE metrics, which tend to measure token overlap -  since the dataset is synthetic, this will bias results towards giving a higher score to the model that generated synthetic responses (LLAMA)
- There is one experiment for persona vs no-persona responses but no experiment regarding adaptive memory and no experiment regarding correctly identifying speakers multi-speaker environments.

**Questions:**

Could you please provide more details on how the LLAMA model was fine-tuned and evaluated?
How does adaptive memory work in your system? What is the impact of using dynamic memory vs no memory?

---

### Official Review · Reviewer_fHka · 2025-10-30

**Soundness:** 2
**Presentation:** 2
**Contribution:** 2
**Rating:** 2
**Confidence:** 4

**Summary:**

This paper proposes a dialogue agent designed to support user-specific, long-term interactions across multiple users. To achieve this, the system maintains a user memory bank that stores user-specific history. For each user, the bank stores a summarized long-term memory and the complete most recent ten conversations, along with the extracted user persona. The model employs an off-the-shelf speaker identification module to distinguish among users and, accordingly, construct or access each user’s memory and persona. Dialogue history summarization and persona extraction are implemented using an off-the-shelf commercial LLM (GPT-4o).

For response generation, the agent incorporates both the user-specific persona and relevant historical information retrieved from the corresponding memory. To further enhance performance, the authors introduce a persona-oriented dataset named Personalized Agent chaT (PAT), curated using large language models (GPT-4o and LLaMA-405B). They construct diverse personas by extracting and extending personality traits from an existing dataset (MSC) and automatically generate question–answer pairs under LLM-generated scenarios. Experimental results show that an LLaMA-70B model fine-tuned on PAT outperforms several proprietary models on the PAT test set.

**Strengths:**

i. The idea of maintaining a user-specific memory bank for long-term, personalized dialogue has clear practical application value. The paper’s effort to implement such a system is meaningful.

ii. The curation of a new persona dataset generated under diverse scenarios provides a potentially useful resource for future studies.

**Weaknesses:**

i. In its current form, the paper does not meet the quality standards required for publication due to significant issues in writing, formatting, and overall presentation. These include:

- References are not properly formatted. Citations should be enclosed in parentheses, but are currently inconsistent throughout.

- Lines 116–125 are a paraphrased version of lines 98–107. This repetition appears to be an editing mistake.

- Table 3, which presents the main experimental results, is incomplete, with several columns missing, and it exceeds the confined text area.

- There is a reference typo in line 215 that should be corrected.

ii. Lack of novelty and, in its current form, reads more like an implementation report rather than a research paper.

Although the authors present a new dataset, there is little to suggest an original contribution to the field in terms of system design or methodology. They claim their system is long-term and user-specific. However, summarization of dialogue as long-term memory is not new [1]. For user-specific, they just use an off-the-shelf speaker identification module to distinguish among users and maintain a user database. This approach seems more like an engineering solution rather than a novel research contribution.

iii. Lack of Alignment with Main Motivation and Insufficient Evaluation

**Questions:**

While the curation of a new persona dataset under diverse scenarios is a positive contribution, its relevance to the paper's core motivation—“building a long-term, user-specific system”—is unclear. The dataset does not include long-term dialogue history or multi-user scenarios, which are essential components of the paper's claimed objective. Instead, the dataset primarily consists of common persona data with a variety of personas and persona-based dialogues, which may not fully support the goal of a long-term, user-specific system.

Furthermore, the experiments do not adequately evaluate the quality and effectiveness of the dataset. The models were tested on the in-distribution dataset, and the performance of the fine-tuned models is unsurprising. A model fine-tuned specifically on the training data will naturally outperform a zero-shot model. Additionally, using metrics like BLEU or ROUGE can only evaluate the model’s ability to mimic language patterns in the training data rather than its ability to capture meaningful, long-term user interaction. Thus, the evaluation lacks a clear demonstration of the dataset’s utility in the context of the paper's proposed system.

[1] https://arxiv.org/abs/2308.15022

---

### Official Review · Reviewer_Gat5 · 2025-10-30

**Soundness:** 1
**Presentation:** 2
**Contribution:** 1
**Rating:** 2
**Confidence:** 5

**Summary:**

This paper presents two core contributions: (1) the Adaptive Friend Agent (AFA), a retrieval-augmented generation (RAG) pipeline that produces personalized dialogues, and (2) a persona-rich dataset bootstrapped from MSC.

AFA identifies each interlocutor by speaker ID, compresses the last 10 turns into a concise memory query with GPT-4o, retrieves relevant historical snippets from an external store, and fuses them into the live context.

To build the dataset, the authors extract speaker personas from the raw conversations, instantiate them in diverse interaction scenarios, prompt GPT-4o to generate new dialogues, and automatically validate the results.

Fine-tuning Llama-70B on this corpus yields responses that surpass those of closed-source LLMs on BLEU and ROUGE, demonstrating the value of both the RAG pipeline and the curated data.

**Strengths:**

1. This paper curated a mutli-turn dialogue dataset.

**Weaknesses:**

1. The approach lacks novelty: RAG, memory mechanisms, and LLM-synthetic data are already standard in both research and industrial settings.
2. Experiments rely solely on BLEU/ROUGE metrics that are now considered inadequate for assessing dialogue quality, coherence, or persona consistency.
3. Result analysis is minimal; the paper offers almost no ablation, error inspection, or discussion of what actually drives “personalized” generation.
4. Speaker ID is introduced but never justified; simply tagging utterances with an ID adds no textual personalization and falls outside the paper’s stated focus on content-level adaptation.
5. The appendix reference on L215 is broken, and the Table on L368 exceeds page width.

**Questions:**

N/A

---

### Official Review · Reviewer_nKcV · 2025-10-30

**Soundness:** 2
**Presentation:** 1
**Contribution:** 2
**Rating:** 2
**Confidence:** 4

**Summary:**

The paper proposes Adaptive Friend Agent, a modular framework integrating speaker recognition, personalized memory, and LLM-based dialogue generation to support multi-user conversational systems.
It also introduces a synthetic dataset, Personalized Agent chaT, for model training and evaluation.
The paper claim that AFA, fine-tuned from LLaMA-70B, surpasses commercial and ablation baselines on BLEU/ROUGE metrics, and that memory and speaker identification are key contributors to its performance.

**Strengths:**

1. The system targets realistic multi-user environments (e.g., families, shared devices), combining identity detection and personalized response.
2. The topic is very interesting and meaningful.
3. The paper clearly describes four components: speaker identification, dynamic user profiles, persona synchronizer, and adaptive response generator, supported by detailed figures and workflow diagrams.
4. Table 3 shows measurable BLEU/ROUGE gains when enabling adaptive persona fine-tuning for LLaMA-70B.

**Weaknesses:**

1. Both fine-tuning and evaluation rely on the synthetic PAT dataset generated by LLMs. This raises serious data leakage and overfitting risks, the model may simply learn to mimic the synthetic data distribution rather than generalize.
2. The architecture is an incremental combination of known components: pretrained speaker recognition, top-k semantic memory retrieval, and LLM generation. There is no new algorithmic innovation or theoretical insight separating AFA from prior “memory-augmented personalized LLM” work.
3. The paper promises “Our code at Link” but provides no actual URL.

**Questions:**

1. Do you have real human multi-speaker audio dialogues for evaluation? What is the actual speaker recognition error rate and its downstream effect?
2. When will code, models, and data be released?

---

### Official Review · Reviewer_Dtqe · 2025-11-01

**Soundness:** 3
**Presentation:** 3
**Contribution:** 2
**Rating:** 4
**Confidence:** 4

**Summary:**

The paper proposes Adaptive Friend Agent (AFA), a modular conversational method for multi-user personalization. AFA uses (i) speaker identification (SpeechBrain embeddings) to route utterances to a user, (ii) a Dynamic User Profile Store with short-term “temporary” buffers and long-term “permanent” summaries, (iii) a Persona Synchronizer that continually updates user traits from interactions, and (iv) an Adaptive Response Generator (LLM) conditioned on retrieved, per-user memories.
To support training/evaluation, the authors introduce PAT, a synthetic dataset of 50k persona-grounded query–response pairs across 12 everyday scenarios (e.g., travel, shopping, emotional support).

**Strengths:**

The paper presents a clear pipeline (speaker ID → memory → persona update → adaptive generation), which is conceptually coherent and easy to follow. This makes the system replicable and highlights the interaction between perception and personalization components.

The dual-layered memory (temporary and permanent) is a thoughtful design that balances short-term contextual relevance with long-term user modeling. This mirrors how memory compression is treated in recent personalized dialogue agents.


The introduction of PAT significantly enriches evaluation resources for persona-grounded dialogue. Its multi-domain coverage and persona variety demonstrate commendable effort in dataset curation and scalability. Large synthetic corpus spanning 12 scenarios and 133 personas, with basic LLM-as-judge validation.

**Weaknesses:**

The paper positions AFA against: (1) multi-session personalization (MSC) and memory-augmented dialogue (MemoryBank, MemoChat), and (2) persona-based chat (PersonaChat, RAP), arguing novelty in simultaneous multi-user handling via speaker recognition plus per-user memory stores that evolve over time.  The framing is reasonable; however, the empirical comparison is largely intra-system (AFA variants) and across base models rather than against recent multi-user assistants beyond citations, limiting external validity.


Since both training and testing rely on the PAT dataset generated via LLM prompting, results may reflect distributional alignment rather than genuine generalization. A real-user or human-annotated evaluation is essential for stronger validity.

The evaluation metrics (BLEU, ROUGE, Distinct) primarily measure surface overlap and lexical diversity, not deeper personalization qualities such as factual recall of user traits or persona consistency.

Although the abstract claims that memory and speaker ID modules are critical, the paper lacks detailed ablation studies, statistical variance, and error analyses to substantiate this claim quantitatively.

The paper stores and updates per-user voice embeddings and persona data but does not address data retention policies, consent mechanisms, or safeguards. This is an omission for deployable personalization systems.

**Questions:**

NA

---

### Meta-Review · Area_Chair_vZrb · 2026-01-07

**Summary:**

Across reviews, the main decision-driving concerns are that AFA is framed as a multi-user personalized assistant but is evaluated primarily on an LLM-synthetic, persona QA-style dataset (PAT), using overlap-based metrics (BLEU/ROUGE) and limited baselines.
Reviewers repeatedly question external validity (synthetic train/test distribution alignment, possible leakage/overfitting, and lack of real-user or human-annotated evaluation), novelty (pipeline-level integration of known components), and insufficient experimental substantiation (missing/weak ablations, no speaker-ID error analysis and downstream impact, minimal error analysis, and limited comparisons to recent multi-user personalization systems).

**Reviewer Concerns:**

Addressed by the rebuttal:
- None (no rebuttal was provided).

Still outstanding:
- Evaluation validity / generalization (Dtqe, nKcV, fHka, mNQc): Heavy reliance on LLM-generated PAT for both training and testing; risk of distributional alignment and overfitting; lack of real-user/human evaluation and out-of-distribution testing.
- Metrics inadequacy (Dtqe, Gat5, fHka, mNQc): BLEU/ROUGE/Distinct are not sufficient to assess personalization quality.
- Novelty / contribution (nKcV, Gat5, fHka, mNQc): Perceived as an incremental combination of standard components, with limited algorithmic novelty or insight.
- Missing/weak baselines (All reviewers): Limited comparisons to strong, recent multi-user personalization assistants; evaluation task (persona QA) not clearly aligned with multi-user long-term conversational setting; no rigorous speaker-ID evaluation.
- Ablations / analysis (Dtqe, Gat5, mNQc): Insufficient ablation studies, variance, and error analysis to support claims that memory and speaker ID are critical.
- Release / reproducibility (nKcV, Gat5, fHka): Missing promised code URL; broken references/appendix issues.

**Reviewer Scores:**

Scores unchanged because no rebuttal was provided.

---

### Decision · Program_Chairs · 2026-01-26

Reject